# Shoe feature recommendations for different running levels: A Delphi study

**Eric C. Honert**[1☯]*, **Maurice Mohr**[1,2☯], **Wing-Kai Lam**[3,4,5☯], **Sandro Nigg**[1☯]

**1** Human Performance Laboratory, Faculty of Kinesiology, University of Calgary, Calgary, Alberta, Canada, **2** Institue of Sport Science, University of Innsbruck, Innsbruck, Austria, **3** Guangdong Provincial Engineering Technology Research Center for Sports Assistive Devices, Guangzhou Sport University, Guangzhou, China, **4** Department of Kinesiology, Shenyang Sport University, Shenyang, China, **5** Li Ning Sports Science Research Center, Li Ning (China) Sports Goods company, Beijing, China

☯ These authors contributed equally to this work.
* eric.honert@ucalgary.ca

**Data Availability Statement:** All relevant data are within the paper and its Supporting Information files.

## Abstract

Providing runners with footwear that match their functional needs has the potential to improve footwear comfort, enhance running performance and reduce the risk of overuse injuries. It is currently not known how footwear experts make decisions about different shoe features and their properties for runners of different levels. We performed a Delphi study in order to understand: 1) definitions of different runner levels, 2) which footwear features are considered important and 3) how these features should be prescribed for runners of different levels. Experienced academics, journalists, coaches, bloggers and physicians that examine the effects of footwear on running were recruited to participate in three rounds of a Delphi study. Three runner level definitions were refined throughout this study based on expert feedback. Experts were also provided a list of 20 different footwear features. They were asked which features were important and what the properties of those features should be. Twenty-four experts, most with 10+ years of experience, completed all three rounds of this study. These experts came to a consensus for the characteristics of three different running levels. They indicated that 12 of the 20 footwear features initially proposed were important for footwear design. Of these 12 features, experts came to a consensus on how to apply five footwear feature properties for all three different running levels. These features were: upper breathability, forefoot bending stiffness, heel-to-toe drop, torsional bending stiffness and crash pad. Interestingly, the experts were not able to come to a consensus on one of the most researched footwear features, rearfoot midsole hardness. These recommendations can provide a starting point for further biomechanical studies, especially for features that are considered as important, but have not yet been examined experimentally.

## Introduction

Matching running footwear features to the functional needs of the runner has the potential to improve footwear comfort [1,2], enhance running performance [3,4] and reduce the risk of overuse injuries [1,5]. The majority of biomechanical studies have examined the effects of

**Funding:** "Li-Ning provided support in the form of a salary for WKL, but did not have any additional role in the study design, data collection and analysis, decision to publish, or preparation of the manuscript. The specific role of WKL is articulated in the 'author contributions' section".

**Competing interests:** "WKL affiliation with Li-Ning does not alter our adherence to PLOS ONE policies on sharing data and materials".

footwear interventions for a general group of runners and/or athletes rather than specific groups of runners, stratified according to their training status and/or running experience. This is despite evidence that runners of different levels (e.g. novice, recreational, high caliber) have clear differences in functional needs and running goals that need to be addressed in the design of their footwear (e.g. through cushioning or stability features, [6–9]). As a result, there is a large gap of knowledge on how to match specific footwear features, and their properties, to runners from different levels. This gap in knowledge limits the potential beneficial effects that more individualized footwear may have on comfort, performance or injury risk.

Literature has presented a variety of definitions for different running levels. Studies have suggested standard definitions for different runner levels, which have been derived from subjective questionnaires [6,7]. However, these definitions are often not translated to biomechanical studies examining footwear features for runners. For example, subjective questionnaires indicate that recreational runners run, on average, between 25 and 35 km/week [7]. Yet, biomechanical studies have recruited "recreationally running" subjects with an average training distance between 10 km/week [10] and 50 km/week [11]. On the other hand, literature has consistently described novice runners as having little to no running experience in the past year (see [9] for a Meta-Analysis of novice runners). Due to the wide range of definitions for running levels used in literature, there is a need to reach a consensus on an operational definition for different running levels.

Modern running shoes are complex systems. They incorporate many different features (e.g. crash-pads, heel counters, flares, midsole hardness) and each of these features can be included, excluded and/or tuned individually to modify the characteristics of the final running shoe system (e.g. cushioning, stability, heel-to-toe transition, energy return). Some of these shoe features have been studied more extensively than others [12,13]. A strong research focus on certain footwear features does not necessarily translate into agreement on how modifying these features may affect the running mechanics, performance, injury risk or footwear comfort in runners of different levels. For example, a recent review found inconclusive evidence regarding the biomechanical effects of different midsole hardness—one of the most studied footwear features [13]. On the other hand, there has been little scientific attention on footwear features such as outsole traction or forefoot flares. A lack of scientific attention could indicate that the prescription of these features to different runner levels is trivial, these features are not considered important by footwear professionals or little is known on how to prescribe these features. An understanding of how footwear experts make decisions about different footwear features and their properties can be obtained through gathering and summarizing opinions of experts in the field of running biomechanics and footwear using a Delphi study. The Delphi method has been utilized for gathering and summarizing opinions via survey-based responses of an expert panel in order to obtain consensus on complex topics. For example, this technique has been successfully applied to establish the now frequently reported "Minimalist Index" of running shoes [14]. Such an understanding can target future systematic investigations around the presumed optimal property of important footwear features.

The purpose of this study was to utilize a Delphi technique to summarize the opinions of running footwear experts and reach consensus on 1) runner level definitions, 2) which footwear features are important when designing footwear for different running levels, and 3) matching the specific properties of footwear features to the respective running levels.

## Methods

Footwear experts were asked to complete three rounds of a Delphi study, with each successive round building on the results gathered from the previous round. Three runner level definitions

were refined throughout the three rounds of the Delphi study through expert feedback. Experts were also provided a list of 20 different footwear features. Through the three rounds of the study, experts provided opinions on which features were important and what their properties should be for the three different running levels.

## Delphi study

In total, 142 experts from 18 countries were contacted by e-mail to participate in this Delphi study: 44 academics, 35 journalists, 25 coaches, 24 scientists in the footwear industry, seven bloggers and seven physicians. The participants for this Delphi study were compiled from: authors that appeared on multiple papers from a recent literature review [13], podium presenters at the 2019 Footwear Biomechanics Symposium, coaches of national and/or college track and field teams with publicly available e-mail addresses, scientists working in research and development at the leading running footwear brands, running shoe bloggers and journalists identified from an online search of popular running blogs and magazines and running and/or footwear journalists that Professor Benno Nigg has compiled over the years. All potential participants were contacted via e-mail to participate in this Delphi study. Participants were excluded if they had under two years of experience related to running footwear in their respective fields of expertise. Each participant was provided an implied consent form stating that returning the survey was their agreement to participate. The protocol was approved by the University of Calgary's Conjoint Heath Research Ethics Board (REB19-0240). The footwear experts completed web-based surveys through QuestionPro (questionpro.com) and could provide feedback after the completion of each round of this Delphi study. The participants that completed the first-round survey were invited to participate in the second-round. Similarly, the participants that completed the second-round survey were invited to participate in the third round. To prevent bias in the responses and feedback, all participants' survey responses were anonymized by the QuestionPro platform. All participants were encouraged to e-mail the authors upon completion of each respective round of the Delphi study for additional feedback and/or comments, and to create a list of respondents for successive rounds of the survey.

## Running levels

Three different running levels were initially proposed: novice, recreational and high caliber. The initial characteristics of each running level (Table 1) were defined based on running literature [6,7,9–11,15–20]. The proposed characteristics provide guidelines for runner classification. As such, there were overlaps in the running distance per week between the running levels in order to accommodate runners that train less and have a better running performance. Feedback on the running level definitions was requested from the participants during each round of the Delphi study. The feedback from rounds one and two was integrated into the running level definitions and presented to the participants in rounds two and three, respectively. In each round, the experts rated the running level definitions on a 10-point scale where "1" indicated that the definitions were "Not at all appropriate" and "10" indicated "Most Appropriate".

 **Novice runners—Initial definition.** Novice or occasional runners have little running experience. These runners typically have less than six months of cumulative regular running training (i.e. at least one day per week) over the previous 12 months [9,15,17]. They run zero to three times per week with a maximum of about 20 km per week [6,7,10]. Novice runner performance (Table 1) was extrapolated from an average running pace [10]. These runners are typically not involved in marathons [7]. Surveys have shown that these runners run to improve general health, manage stress and weight [7]. Novice runners may choose footwear based on comfort [16], reduce injury risk and improve performance [7].

**Table 1. Initial definitions of running levels.**

| | Level 1<br>Novice | Level 2<br>Recreational | Level 3<br>High-caliber |
|---|---|---|---|
| **Running experience** | Less than six months of regular* running experience | More than six months of regular* running experience | More than three years of regular* running experience |
| **Running habits** | 0–3 sessions / week | 1–5 sessions / week | > 3 sessions / week |
| | 5–20 km / week | 15–50 km / week | > 30 km / week |
| **Running performance** (times are for male runners) | 5km time > 30 min *OR* | 5km time > 20 min *OR* | 5 km time 15–20 min$ *OR* |
| | 10km time > 60 min | 10km time > 45 min *OR* | 10 km time 30–45 min$ *OR* |
| | No marathon racing | Marathon time 3–4.5 h | Marathon time <3h$ |
| **Running motivation** (ordered according to importance) | Improve general health | Improve general health | Improve general health |
| | Stress management | Stress management | Stress management |
| | Weight management | Team affiliation | Competition |
| **Priorities for footwear design** (from high to low) | 1) Improve comfort | 1) Improve comfort | 1) Improve performance |
| | 2) Reduce injury risk | 2) Reduce injury risk | 2) Improve comfort |
| | 3) Improve performance | 3) Improve performance | 3) Reduce injury risk |

The (*) indicates regular running experience defined as running at least once per week. The ($) indicates that elite runners with faster race times than high caliber runners were not considered since they represent a small percentage of the population and may require individual running footwear recommendations.

**Recreational runners—Initial definition.** The recreational group is the largest running group [7]. These runners typically have more than six months of cumulative regular running training (i.e. at least one day per week) over the previous 12 months [10,15]. They run one to five days per week for a total of 10 to 50 km per week [6,7,10,11,15]. Recreational running performance (Table 1) was extrapolated from running times reported in [21]. Surveys have shown that these runners run to improve general health, manage stress and be involved with a team [7]. Recreational runners may choose footwear based on comfort [16], reduce injury risk and improve performance [7].

**High caliber runners—Initial definition.** High caliber runners have significant distance running experience, train almost daily and regularly compete in regional to international competitions [18]. These runners typically have over three years of regular running experience [7,20]. They run about three times per week for at least 30 km per week [6,7]. High caliber running performance (Table 1) inclusion criteria has been reported in several running studies [18–20]. Surveys have shown that these runners run to improve general health, manage stress and compete [7]. High caliber runners may choose footwear based on performance, comfort and reduced injury risk [7,16].

## Footwear features

Twenty running footwear features were initially assessed in this Delphi study. These features were chosen from an initial list of 31 footwear features that were identified based on a preliminary literature review, market analysis and internal discussion. Two influential studies during this process were reports from [6] and [14]. This initial list was reduced to 23 features by removing or joining related features that were reflected in other features or similar in their function, respectively (e.g. remove midfoot midsole hardness and only retain forefoot and rearfoot midsole hardness). Pilot testing with four footwear experts (not included in the main study) indicated that a survey including 23 features required more than an hour to complete and could potentially lead to a high-drop out rate. Therefore, we limited the number of footwear features to 20, by removing features that pilot participants indicated had low relevance

(e.g. upper overlays or varus alignment). In return, the option was added for experts to suggest footwear features that should be added to the questionnaire. The final 20 footwear features assessed in this Delphi study were (see S1 Appendix for description of each feature): crash pad, forefoot flares, forefoot longitudinal bending stiffness, forefoot midsole hardness, heel counter, heel flare, heel (stack) height, heel-to-toe drop, insole shape, medial post, midfoot longitudinal bending stiffness, midsole thickness, outsole traction, rearfoot midsole hardness, rocker (heel), shoe mass, toe spring (forefoot rocker), torsional bending stiffness, upper material (breathability) and upper material (elasticity).

The importance of the footwear features was assessed in the first-round and verified in the second-round. In the first-round, participants were asked if each footwear feature was important when designing footwear for different running levels. The experts could choose between the following for each footwear feature: (a) is important, (b) is not important or (c) they do not know if it is important. If over 75% (a similar threshold to [22,23]) of the first-round participants selected option (a), the footwear feature was defined as important. The important features were then presented to the second-round participants. The participants were asked if they agreed with the list of the features selected as important/non important on a 10-point scale where "1" indicated that the list of important/non important features was "Not at all appropriate" and "10" indicated "Most Appropriate". The list of important features was verified if over 75% of the second-round participants answered with a seven or higher on the 10 point-scale. The second- and third-rounds of the Delphi study were then limited to the important footwear features. In each round, the experts were asked if other footwear features should be included in the Delphi study. If there were at least five suggestions to add a certain feature, this new feature was added to the subsequent round. The participants were then asked if this new feature was important.

## Footwear feature properties

The experts were asked to recommend footwear feature properties for the different running levels in each round of the study from a multiple-choice selection (see S1 Appendix for the lists of footwear feature properties). Most footwear feature properties were defined based on the reviewed footwear literature (see S1 Appendix). If there was no related literature (e.g. upper elasticity), properties were provided based on commercially available shoes. In rounds 2 and 3, the results from the previous round were presented to the participants. If at least 51% of the participants agreed on a footwear feature property (a similar threshold to [24]) for a specific running level (e.g. high breathability for novice runners), the participants would be asked if they agreed with the consensus the next round. If at least 51% of the participants verified the consensus, the experts were not asked again to recommend a footwear feature property for that running level (see Fig 1). In comparison to the consensus for the importance of shoe features (75%), the threshold for consensus was set lower for agreement on footwear feature properties (51%) because of the greater number of available response options.

## Additional Delphi questions

In the second-round of the Delphi study, we aimed to quantify why the participants chose "I don't know" for the footwear feature properties. The participants were prompted to choose one of the following if they selected "I don't know": feature is not well defined, feature is dependent on foot contact pattern (e.g. heel strike), feature is dependent on biomechanical variables (e.g. foot inversion), feature has interplaying effects with other shoe features, feature function is not known or other. These questions were included due to a

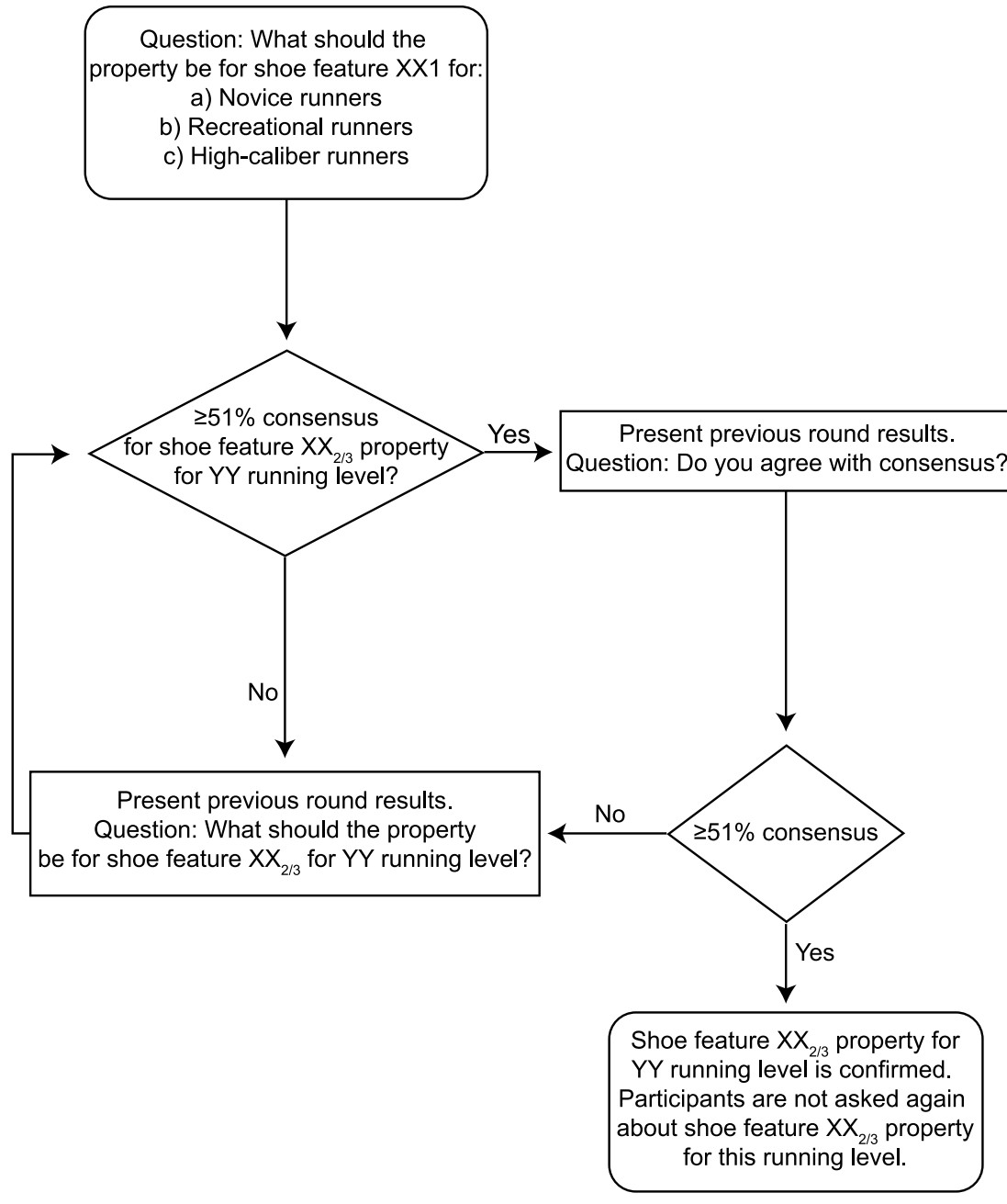

**Fig 1. Flowchart describing the consensus and verifying consensus process for different shoe feature properties (XX) for each running level (YY).** The participants were asked to provide feedback for the recommended properties for all runner levels on all 20 shoe features ($XX_1$). In the second- and third-rounds, the participants were asked to provide feedback for the recommended properties for all runner levels on the important shoe features and any additional shoe features the participants recommended ($XX_{2/3}$).

high frequency of "I don't know" responses for some footwear feature properties. These questions were only included in the second-round as we received feedback that the questionnaire was time consuming, which may have increased drop-out rate if included in the third-round.

## Statistical analysis and visualization

Paired statistical analyses were performed to determine if the running level definitions improved through the three rounds of this Delphi study. A Friedman's test was performed utilizing the subjective ratings from the respondents that participated in all three rounds of the study ($N = 24$). If the Friedman's test revealed a significant effect, follow-up Wilcoxon signed-rank tests with a Bonferroni correction were performed to investigate pairwise differences between the individual rounds. The significance level α was set to 0.05 for all statistical tests. The median and inter-quartile ranges of the participants' responses were also computed from the subjective ratings. These descriptive statistics were computed to demonstrate if the ratings increased and if there was less variability in the responses. All analyses were performed in MATLAB (version 2019a, MathWorks, Natick, MA, USA). Figures were created in MATLAB and Adobe Illustrator (version 22.1, San Jose, CA, USA).

# Results

## Participation

Of the 142 experts initially contacted, 29 responded to the first-round of this Delphi study (Fig 2, Table 2). Twenty-five respondents participated in the second-round and 24 participated in the third-round (Fig 2, Table 2). Note that one academic moved to industry from academia between rounds one and two.

## Running level definitions

The respondents' rating of the running level definitions improved as the Delphi study progressed, $\chi^2$ (2, $N = 24$) = 13.95, $p = 0.0009$. The median rating increased each round and the

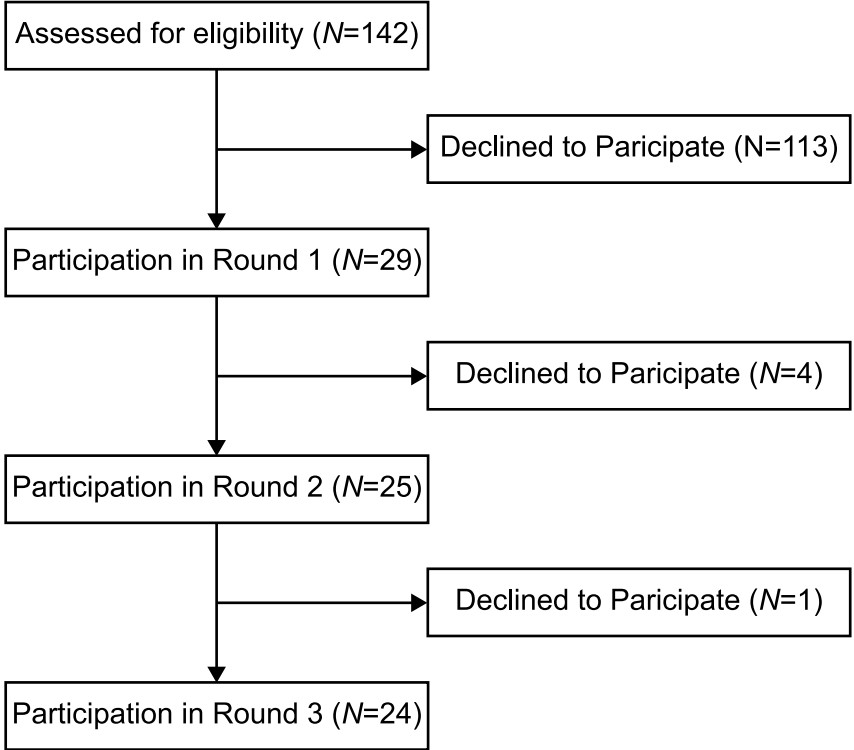

**Fig 2. Participation in each round of this Delphi study.**

Table 2. Number of participants and their experience investigating/designing footwear.

| | Experience (yrs) | Round 1 | Round 2 | Round 3 |
|---|---|---|---|---|
| Academic | 2–5 | 1 | 0 | 0 |
| | 5–10 | 6 | 4 | 4 |
| | 10+ | 8 | 7 | 7 |
| Professional in the footwear industry | 2–5 | 0 | 1 | 1 |
| | 5–10 | 2 | 2 | 2 |
| | 10+ | 8 | 8 | 8 |
| Clinician | 10+ | 2 | 1 | 1 |
| Journalist | 5–10 | 1 | 1 | 0 |
| Coach | 10+ | 1 | 1 | 1 |
| Total | | 29 | 25 | 24 |

Note that one academic moved to industry between the first and second rounds of this study.

interquartile range decreased. For example, 69% of respondents rated the running level definitions between 7 and 10 in the first-round which increased to 88% of respondents in the third-round (see Fig 3). The increase in the running level scores between the first and third rounds was statistically significant ($p = 0.006$). The increased running level ratings were accompanied by changes to the running level definitions. The changes to the "novice" running level definition for the second-round were: increased running experience to one year and replaced "stress management" with "enjoyment" for running motivation. The changes to the "recreational" running level definition for the second-round were: increased running experience to greater than one year and replaced "stress management" with "enjoyment" for running motivation. The changes to the "high-caliber" running level definition for the second-round were: increased running habits to >4 sessions/week and >50 km/week, replaced "stress management" with "enjoyment" for running motivation, re-order the running motivation to 1) Competition, 2) Improve general heath, and 3) Enjoyment, and re-order the priorities for footwear design to 1) Improve performance, 2) Reduce injury risk, 3) Improve comfort. We also specified the running performance as males between the ages of 18 to 34. Subsequent changes to the running level definitions were to ensure that the high caliber and recreational runner 5 km and 10 km times were indicative of the respective marathon times. These updates resulted in the final runner level definitions in Table 3.

## Footwear features

Twelve of the 20 footwear features reached the level of consensus to be considered important. The majority (92%) of the second-round respondents rated the appropriateness of the 12 important footwear features as a 7/10 or higher. "Lacing system" was added to the second-round of this Delphi study as five first-round respondents suggested that it should be included in the list of footwear features. This feature did not reach the threshold of consensus in the second-round (68%, Table 4) to be considered important. "Toe spring" was initially not an important footwear feature as only 19/29 (66%, Table 4) first-round respondents thought it was important for footwear design. Five second-round participants suggested to add "toe spring" back into the survey (as it was removed because it was below the threshold of consensus) and 22/24 (92%, Table 4) third-round participants thought that it was important for footwear design.

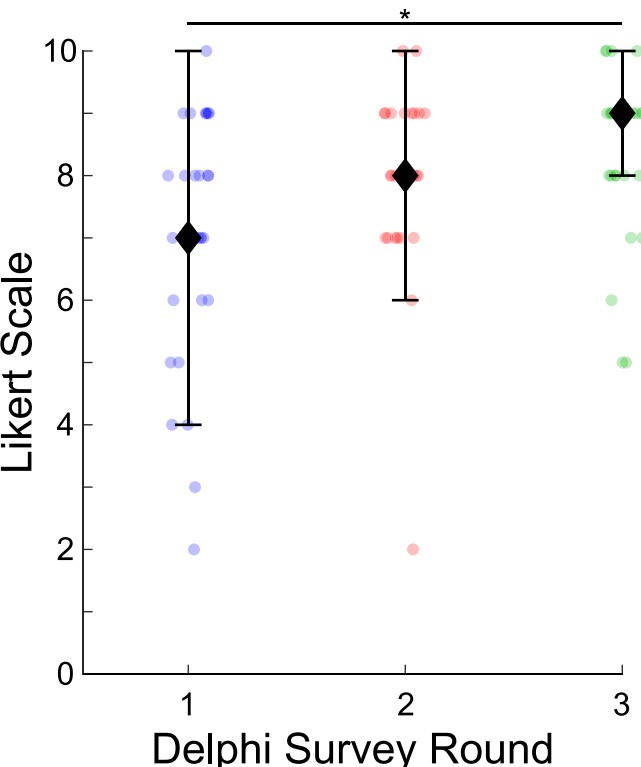

**Fig 3. Subjective rating of the running level definitions for the three rounds of this Delphi study.** Changes in subjective ratings were accompanied by updating the running levels definition based on respondents' feedback. The diamonds represent the median of each round and the bars indicate the interquartile range. Each shaded dot indicates one response made by a respondent. The asterisk (*) indicates a statistical difference in the subjective ratings ($p = 0.006$).

**Table 3. Final running level definitions.**

| | Level 1 Novice | Level 2 Recreational | Level 3 High caliber |
|---|---|---|---|
| **Running experience** | **Less than one year of regular* running experience** | **More than one year of regular* running experience** | More than three years of regular* running experience |
| **Running habits** | 0–3 sessions / week | 1–5 sessions / week | **> 4 sessions / week** |
| | 5–20 km / week | 15–50 km / week | **> 50 km / week** |
| **Running performance** (example times are for male runners age 18–34) | 5km time > 30 min *OR* | **5km time > 21 min *OR*** | **5 km time 15–20 min$^\$$ *OR*** |
| | 10km time > 60 min | **10km time > 42 min *OR*** | **10 km time 30–42 min$^\$$ *OR*** |
| | No marathon racing | Marathon time 3–4.5 h | Marathon time <3h$^\$$ |
| **Running motivation** (ordered according to importance) | Improve general health | Improve general health | **Competition** |
| | **Enjoyment** | **Enjoyment** | **Improve general health** |
| | Weight management | Team affiliation | **Enjoyment** |
| **Priorities for footwear design** (from high to low) | 1) Improve comfort | 1) Improve comfort | 1) Improve performance |
| | 2) Reduce injury risk | 2) Reduce injury risk | 2) **Reduce injury risk** |
| | 3) Improve performance | 3) Improve performance | 3) **Improve comfort** |

These definitions were refined by the Delphi study participants through the three rounds of feedback. The (*) indicates regular running experience defined as running at least once per week. The ($^\$$) indicates that elite runners with faster race times than high caliber runners were not considered since they represent a small percentage of the population and may require individual running footwear recommendations. Bolded characteristics indicate characteristics that changed from the first characteristics presented to the respondents (Table 1).

**Table 4. Percent of participants that agreed upon the importance of shoe features.**

| Shoe Feature | % Participants | Shoe Feature | % Participants |
|---|---|---|---|
| **Shoe Mass** | 100 | **Toe Spring** | 66/92 |
| **Upper Breathability** | 97 | Heel Counter | 72 |
| **Forefoot Midsole Hardness** | 93 | Medial Post | 72 |
| **Rearfoot Midsole Hardness** | 93 | Midfoot Bending Stiffness | 72 |
| **Heel (stack) Height** | 90 | Upper Elasticity | 72 |
| **Midsole Thickness** | 86 | Insole Shape | 69 |
| **Forefoot Bending Stiffness** | 83 | Lacing System | 68 |
| **Outsole Traction** | 83 | Rocker | 59 |
| **Heel-to-Toe Drop** | 79 | Heel Flares | 55 |
| **Torsional Bending Stiffness** | 79 | Forefoot Flares | 45 |
| **Crash Pad** | 76 | | |

The shoe features with a consensus above 75% were considered important (bolded). The toe spring was initially not considered important (consensus: 66%), but was considered important in the third-round (consensus: 92%). The lacing system was added in the second-round to the study, but was not considered important.

## Footwear feature properties

Twenty-three of the 36 shoe feature properties (3 running levels x 12 important shoe features) reached the 51% consensus threshold (Table 5). Consensus was obtained for upper breathability, heel-to-toe drop, forefoot bending stiffness, crash pad and torsional bending stiffness for all three running levels (Table 5). The consensus for the feature properties from the first- and second-rounds was verified in the second- and third-rounds, respectively (Table 5). There was no consensus for the properties of the toe spring as well as the rearfoot and forefoot midsole hardness for any of the running levels (Table 5). The most frequent response regarding forefoot and rearfoot midsole hardness was "I don't know". In the second-round when participants were asked further about this response, the most frequent answer (4/10 participants) for the forefoot midsole hardness was "feature function is not known". The responses for the rearfoot midsole hardness were spread across the six different responses (see Methods: Additional Delphi Questions for full list of possible responses).

## Discussion

This study provides a unique perspective of footwear experts, most of whom have been examining this topic for 10+ years. These experts indicated that 12 of the 21 footwear features were important for footwear design with respect to different running levels. Experts came to a consensus on the properties for five footwear features for all three running levels. Furthermore, this study has highlighted footwear features that experts consider important but have received little scientific attention, such as: upper breathability, forefoot bending stiffness, heel-to-toe drop, torsional bending stiffness and crash pad (Fig 4). Future, novel research can be performed with these features to add to the collective knowledge of how footwear features can affect the running biomechanics of runners from different levels.

Interestingly, participants in this Delphi study did not come to a consensus for the recommended footwear properties for some of the most researched shoe features: forefoot and rearfoot midsole hardness [12,13]. Previous research has shown that a softer rearfoot midsole can reduce ground reaction force loading metrics such as vertical loading rate or peak impact forces [25–27], which have been hypothesized to reduce running-related injuries [28,29]. The causal relationship between ground reaction force loading metrics and running-related injuries has not been established. Furthermore, examining prospective running injury studies

**Table 5. Shoe feature properties that were most frequently chosen for each running level.**

| Shoe Feature | Running Level | Recommended Property | Round | % Participants | % Participants in agreement with consensus |
|---|---|---|---|---|---|
| Shoe Mass | Novice | 225–275 g | 3 | 43 | - |
| | Recreational | 225–275 g | 3 | 54 | - |
| | High Caliber | <175 g | 1 | 59 | 72 |
| Upper Breathability | Novice | High Breathability | 1 | 69 | 100 |
| | Recreational | High Breathability | 1 | 79 | 100 |
| | High Caliber | High Breathability | 1 | 86 | 100 |
| Forefoot Midsole Hardness | Novice | I don't know | 3 | 50 | - |
| | Recreational | I don't know | 3 | 50 | - |
| | High Caliber | I don't know | 3 | 42 | - |
| Rearfoot Midsole Hardness | Novice | I don't know | 3 | 42 | - |
| | Recreational | I don't know | 3 | 42 | - |
| | High Caliber | I don't know | 2 | 48 | - |
| Heel (stack) Height | Novice | 14–32 mm | 2 | 72 | 88 |
| | Recreational | 14–32 mm | 1 | 65 | 88 |
| | High Caliber | 14–32 mm | 3 | 42 | - |
| Midsole Thickness | Novice | 10–15 mm | 2 | 60 | 58 |
| | Recreational | 10–15 mm | 2 | 52 | 71 |
| | High Caliber | 10–15 mm | 3 | 50 | - |
| Forefoot Bending Stiffness | Novice | Low Stiffness | 1 | 55 | 64 |
| | Recreational | Medium Stiffness | 1 | 66 | 100 |
| | High Caliber | High Stiffness | 1 | 55 | 84 |
| Outsole Traction | Novice | Medium Traction | 1 | 52 | 76 |
| | Recreational | Medium Traction | 1 | 55 | 72 |
| | High Caliber | Medium Traction | 3 | 50 | - |
| Heel-to-Toe Drop | Novice | 8–12 mm | 2 | 56 | 88 |
| | Recreational | 8–12 mm | 3 | 58 | - |
| | High Caliber | 4–8 mm | 3 | 71 | - |
| Torsional Bending Stiffness | Novice | Medium Stiffness | 2 | 72 | 92 |
| | Recreational | Medium Stiffness | 1 | 52 | 76 |
| | High Caliber | Medium Stiffness | 2 | 52 | 88 |
| Crash Pad | Novice | Include Crash Pad | 1 | 76 | 88 |
| | Recreational | Include Crash Pad | 1 | 72 | 88 |
| | High Caliber | Include Crash Pad | 3 | 58 | - |
| Toe Spring | Novice | Mid (16–30 deg) | 1 | 34 | - |
| | Recreational | Mid (16–30 deg) | 3 | 38 | - |
| | High Caliber | I don't know | 1 | 34 | - |

"Round" indicates which round of the Delphi study provided the highest consensus. The footwear feature properties that were above the consensus threshold for Rounds 1 and 2 were all verified in the subsequent rounds as indicated by the percent agreed with consensus (last column).

together demonstrates that ground reaction force loading metrics are not related to injuries [30–40]. This paradigm shift could be the reason for the high frequency of "I don't know" responses for the recommended properties for the forefoot and rearfoot midsole hardness, with the most frequent feedback being "the feature function is not known". Additionally, shoe midsole hardness may interplay with other shoe features such as heel (stack) height or heel-to-toe drop to affect the overall shoe cushioning. This interplay could be the reason for inconsistent findings across studies examining midsole hardness [25,41,42]. In total, further

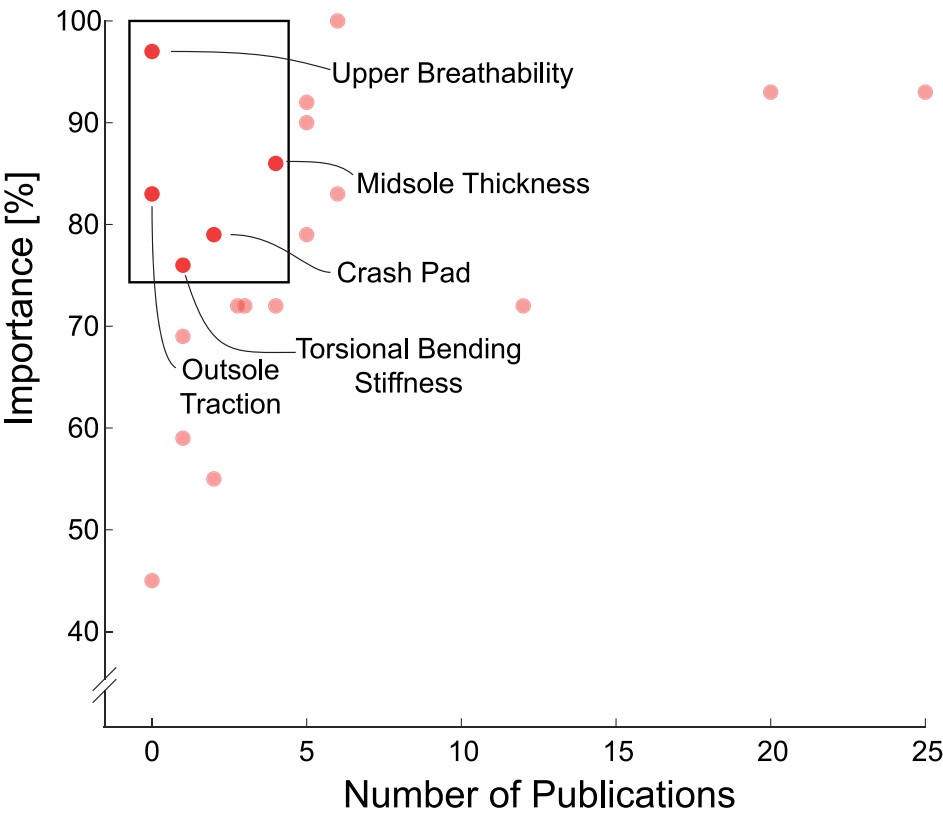

**Fig 4. Footwear feature importance and the number of related publications.** Footwear feature importance as rated by experts in this study in comparison to the number of available publications for each footwear feature based on a recent literature review (with permission from [13]). The footwear features inside the box represent opportunities for future footwear research: while these features were deemed important by footwear experts, only few publications exist regarding how these features affect runners from different levels.

investigations are warranted to determine the biomechanical function of the midsole hardness during running and its relationship with running-related injuries. To achieve this goal, future studies should focus on how footwear properties affect the internal forces (e.g. muscle, tendon, or bone forces) that act on the structures at risk of injury during running [8,43].

Though the experts did provide opinions regarding property ranges for different footwear features, there should be considerations for how these features affect runners and how these features interact. Studies have shown that subject-specific tuning of the forefoot longitudinal bending stiffness can improve running performance [4,44]. Utilizing the expert opinions for groups of runners may overlook this aspect that might be a consideration for footwear design. On the other hand, tuning of multiple features together (e.g. midsole hardness, longitudinal bending stiffness) can provide benefits across a wide range of runners as exemplified by the Nike Vaporfly [19,45]. Such interplay was not addressed in our study as it would exponentially complicate the survey provided to the participants. However, the respondents had mentioned (in feedback and in responses to the round 2 survey, see the S2 Appendix for full responses) that it is difficult to consider some of these footwear features in isolation.

The footwear experts came to a consensus on the running level definitions through slight adjustments to the initial definitions proposed and derived from literature. We opted to provide initial running level definitions to our expert panel rather than letting the panel formulate the definitions independently. This latter approach would have required additional Delphi

rounds prior to the recommendation of footwear features and their properties. Panel formulated definitions may have resulted in different running level definitions compared to the approach presented here. Different running level definitions could have led to altered footwear feature recommendations. However, the experts' consensus on the running level definitions was in agreement with prior literature. This is exhibited by the novice runner level definition which is similar to a definition created based on subjective running questionnaires [7]. These definitions may be viewed more as guidelines as one footwear expert mentioned that "Even elite athletes perform training runs with different intensities, durations, on different surfaces and so on. For each of these runs they might select a different type of footwear." This comment touches on the competing requirements for running shoes as there may be multiple "correct" shoes for a given running level, especially in the high caliber category.

The Delphi methodology is a useful tool for understanding the current status of a given research area, as understood by experts in the field [46]. As such, results from this study can be leveraged to 1) determine if experts are correct in their assumptions (e.g. high forefoot bending stiffness for high caliber runners), 2) determine important areas of limited research and 3) demonstrate areas where there is a lot of research, but little consensus (e.g. rearfoot midsole hardness). The relatively low drop out rate (17%) in conjunction with the extensive feedback obtained from the respondents via open ended questions provides confidence in our methodological approach. The Delphi methodology appears to be relevant when exploring high level topics related to running, and identifying the areas where further research is required.

There are several limitations to acknowledge with this study. Consensus on the recommended footwear feature properties from the third-round could not be confirmed as there was no fourth-round. We believe that the third-round consensus would have been confirmed as the consensus from the first- and second-rounds were confirmed in the second- and third-rounds, respectively. During the second- and third-rounds of the Delphi study, we aimed to reduce the time it took to complete the survey to limit the drop-out rate. To do so, we eliminated footwear features that were not considered important (consensus below 75%) and eliminated footwear feature properties once they were confirmed. Without such eliminations, a different consensus may have been obtained, but there may have also been a larger drop out rate due to the lengthy and repetitive survey. It is recommended to have a drop out rate of less than 30% [47]. We attained a drop out rate of 17%. Additionally, we did not specify whether the footwear recommendations were for male or female runners. As such, these results may not be generalizable between male and female runners as they show distinct anthropometrics and movement mechanics [48]. These results may also not be generalizable to different running surfaces/terrains as we asked participants to only consider running on a hard surface. Furthermore, the final recommendations may be biased as the majority of experts were male (e.g. 22/26 of the final participants). This expert panel was otherwise diverse as nine countries were represented. The recommended footwear feature properties may have been influenced by a dynamic definition of the runner levels, which changed slightly throughout the study. These changing definitions seemed to have little effect on expert opinions on the footwear feature properties as the verifying consensus level was generally higher than the original consensus level (Table 4, last vs. second-to-last column). We also did not specify to the experts how many of the of the categories a runner must match to be considered a "novice", "recreational" or "high caliber" runner. This may have led to minor variations in expert recommendations. Lastly, the data presented here reflect opinions of experts that have experience with footwear. As such, the findings from this study can serve as a valuable starting point for future systematic biomechanical investigations.

## Conclusion

Footwear experts provided feedback on the effects of different footwear features on running biomechanics across three running levels. These experts also came to a consensus on the characteristics of runners in these different running levels. The footwear experts indicated that 12 of the 21 footwear features were important for footwear design. Of these 12 features, experts were able to come to a consensus for five footwear feature properties for all three running levels. These features were: upper breathability, forefoot bending stiffness, heel-to-toe drop, torsional bending stiffness and crash pad. Interestingly, the experts were not able to come to a consensus for one of the most researched footwear features, i.e. rearfoot midsole hardness. These recommendations can provide a starting point for further biomechanical studies, especially for features that have not yet been examined experimentally, e.g. upper breathability.

## Supporting information

**S1 Appendix. Shoe feature descriptions and properties.**
(DOCX)

**S2 Appendix. Raw data from the Delphi study.**
(XLSX)

## Acknowledgments

We would like to thank all of the participants who gave their time to complete the three rounds of this Delphi study including: Michael Asmussen, Christopher Bishop, Jason Bonacci, Nicholas Delattre, Cedric Morio, Tim Derrick, Ned Frederick, Marlene Giandolini, Allison Gruber, Bryan Heiderscheit, Laurent Malisoux, Sabina Manz, Frank Bichel, Benno Nigg, Max Paquette, Craig Payne, Natsuki Sate, Thorsten Sterzing, Matthieu Trudeau, Steffen Willwacher, Beat Hintermann, and Helen Woo. We would also like to thank Ross Miller regarding discussions about prospective studies examining ground reaction force metrics.

## Author Contributions

**Conceptualization:** Maurice Mohr, Wing-Kai Lam, Sandro Nigg.

**Data curation:** Eric C. Honert, Sandro Nigg.

**Formal analysis:** Eric C. Honert.

**Investigation:** Eric C. Honert.

**Methodology:** Maurice Mohr, Wing-Kai Lam.

**Software:** Sandro Nigg.

**Supervision:** Wing-Kai Lam, Sandro Nigg.

**Validation:** Eric C. Honert.

**Writing – original draft:** Eric C. Honert.

**Writing – review & editing:** Eric C. Honert, Maurice Mohr, Wing-Kai Lam, Sandro Nigg.

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
