## [Decision Letter · Decision Letter 0]

1 Apr 2020

PONE-D-20-07118

Shoe Feature Recommendations for Different Running Levels: A Delphi Study

PLOS ONE

Dear Dr. Honert,

Thank you for your contribution to PLOS ONE. After careful consideration, we feel that it has merit but does not fully meet PLOS ONE’s publication criteria as it currently stands. Therefore, we invite you to submit a revised version of the manuscript that addresses the points raised during the review process.

I believe many of our reviewers comments complement each other and provide helpful suggestions to optimize your paper. Please do your best to incorporate recommendations where applicable. Of particular note, reviewer 3 noted areas of identification, which should be addressed, as well as use of the Holtz et al. paper; please be sure this reference is fully available if you intend to use it. I look forward to seeing your revisions. 

We would appreciate receiving your revised manuscript by May 16 2020 11:59PM. To enhance the reproducibility of your results, we recommend that if applicable you deposit your laboratory protocols in protocols.io, where a protocol can be assigned its own identifier (DOI) such that it can be cited independently in the future. For instructions see: http://journals.plos.org/plosone/s/submission-guidelines#loc-laboratory-protocols

We look forward to receiving your revised manuscript.

Kind regards,

Chris Harnish, PhD

Academic Editor

PLOS ONE

Journal Requirements:

1. Please provide additional details regarding participant consent. In the ethics statement in the Methods and online submission information, please ensure that you have specified (1) whether consent was informed and (2) what type you obtained (for instance, written or verbal, and if verbal, how it was documented and witnessed). If the need for consent was waived by the ethics committee, please include this information.

Reviewers' comments:

Reviewer's Responses to Questions

**Comments to the Author**

1. Is the manuscript technically sound, and do the data support the conclusions?

Reviewer #1: Yes

Reviewer #2: Yes

Reviewer #3: Yes

2. Has the statistical analysis been performed appropriately and rigorously? 

Reviewer #1: Yes

Reviewer #2: N/A

Reviewer #3: Yes

3. Have the authors made all data underlying the findings in their manuscript fully available?

Reviewer #1: Yes

Reviewer #2: Yes

Reviewer #3: Yes

4. Is the manuscript presented in an intelligible fashion and written in standard English?

Reviewer #1: Yes

Reviewer #2: Yes

Reviewer #3: Yes

5. Review Comments to the Author

Reviewer #1: General Comments:

The reviewer would like to commend the authors for undertaking an important and interesting topic. Determining the shoe recommendations for different running levels is an important topic, that can aid clinicians and running coaches in choosing the right foot wear for different runners of different abilities.

Overall this is a well written manuscript, with good methodology. There are some specific comments which are written below.

Abstract:

General comment: For an abstract, the background should be brief. Suggest only have 2 sentence for the background. I do not think you need to describe why a Delphi study is powerful within the abstract. I think the first 3 sentences would suffice, and potentially reduce these three sentences into 2.

Within the abstract methods, a little bit more information is needed. For example, how many questions did the study begin with, and how were they whittled down through the three rounds, and how was data tallied. Further, within the results, you describe that there were originally 20 proposed variables. This is an example of something that needs to be in the methods.

Need key words at the end of the abstract.

Introduction:

Line 54: Delete the parenthetical citation fully written citation, should just be a reference number.

Line 67: Same here, please deleted written citation, should just be a reference number.

Line 69-70: Reword this to not be a numbered list. Within the intro, it should just be written sentences.

Line 71: You state, “it is close to impossible for running footwear professionals to provide evidence-based recommendations for footwear properties for runners of different levels.” But then you go on to say you are performing a Delphi to find the best recommendations from the experts. I think this is contradictory. I think you should focus more on how there is not clarity on professional recommendations for footwear for different running skills or groups.

Line 73-76: Why are aims here and the purpose in the final introduction paragraph? This is confusing for the reader. Suggest only having the purpose at the last intro paragraph and deleting the aims.

I think the third to last and second to last paragraphs can be amalgamated into one paragraph. Further, the second to last paragraph ends abruptly and a better conclusion is need to set up the purpose paragraph.

Methods:

General comments: An overall study design sub section is needed at the beginning of the methods. This should give the 10,000 foot view of the study.

You need to give inclusion/exclusion criteria for who was considered an expert for this study.

Lines 103-117: I see that 142 experts were contacted. How many responded and were included. A flow chart might help the reader to understand this process.

Line 122: Need to cite the running lit used. Further, it is confusing with the parenthetical statement “as detailed below. Suggest deleting this.

Novice versus Recreational runner definition: In the novice group, you state that they run no more than 20km/week, but in the recreational group, they run 10-50 km/week? How do delineated between someone that runs 15-20 km/week? Is this based off of times per week (0-3 v 1-5)? Please clarify.

High Caliber runners: I see the same thing here, they run 30km+/week. Please clarify

Line 211: Can you explain further why the ‘don’t know’ questions were not included in round 3?

Line 218-219: Please add the software program used to calculate these statistics.

Results:

General comment: It is not recommended to use bullet points within the results. Please edit accordingly

Overall the results are well written.

Discussion:

Line 307-10: These are more article strengths and should be moved to the strength and limitations section, not the summary discussion paragraph.

Line 315-17: This is a future research, implications, and/or conclusion sentence and should be moved

Line 369: Suggest deleting the term ground truth and just state that this should serve as valuable information, etc.

Line 373-381: suggest that this paragraph be moved before the limitations paragraph.

Line 384-386: Delete the first sentence, this can be said in the strengths paragraph but the conclusion paragraph should focus on the findings and future directions.

Reviewer #2: Overall this manuscript fills an obvious void in the literature and aims to assist researchers, clinicians, coaches, and running enthusiasts with shoe prescriptions, while also informing future running shoe research. This work is generally well written and free from fundamental flaws; however, several minor revisions to the proposed article will undoubtedly improve this already great work.

1. The words "the participants" are over utilized throughout the manuscript. Varied diction will help to maintain reader interest and attention.

2. As this is a study employing Delphi techniques no statistical analyses are necessary and furthermore, no analyses were actually conducted. The "Statistical Analysis" section is therefore unnecessary and the subsequent descriptive statistics can simply be presented in the "Results" as well as Fig 3.

3. A more clear and consistent distinction between footwear properties and features throughout the manuscript would improve readability.

4. "Appendix A" utilizes the term "categories" as opposed to "properties" further illustrating the previous point.

5. Additional headings for the "Footwear Properties" in the "Methods" and "Results" sections would assist readers navigating between parts of the manuscript.

6. The described methods for determining footwear features and feature properties importance is challenging to read at times (particularly lines 169-179; lines 188-192); please try to concisely and succinctly explain these steps.

7. Lines 181-184 seem somewhat redundant.

8. The reference to Fig 2 in line 182 seems somewhat premature. Describing the general flow of these methods prior to interpreting Fig 2 made this section easier for this reviewer to understand.

9. It is not clear how the Likert scale used to rate footwear features (as described in the "Methods" section) is actually used in this study.

10. Fig 2 is very helpful, but a threshold of >50% is provided when the text describes using a 51% threshold.

11. While minor, the software used to produce images was not stated.

12. Line 162 - Explicitly cite why/where the 20 features considered comes from.

13. The inclusion of 2 aims and 3 purposes is somewhat confusing. I recommend removing the aims from your "Introduction" as they do not match the "Methods" and "Results" sections as obviously.

14. Please ensure that permissions for any adapted images (i.e. Figs 1 & 4) are provided as necessary.

15. A limitation that seems somewhat overlooked is that the definitions of runner levels changed throughout iterations. As these definitions changed, so too may have respondents' recommended properties. While the 3 repetitions and consensus measures may help to quell these concerns, it seems important to consider the implications of these interconnected moving targets.

16. If possible, I would like to know more about your "Additional Delphi Questions" results in the discussion. I read some of the statements in your raw data set and found the additional insights very compelling. You do a good job of introducing some of the identified themes in your "Discussion" but I feel that a bit more would elevate the current manuscript.

17. Tables 5 and 6 both seem to provide complementary results. Is there a way to combine them or make the more exclusive from one another?

18. Consider a CONSORT diagram so readers can better understand the development of the expert panel round by round.

19. Please expand on how your panel may or may not influence your conclusions in the "Discussion" (e.g. Where they all from the US? Do they disproportionately represent companies with financial interests in designing complicated shoes? Etc.).

20. Please discuss how providing the expert panel with definitions in round 1 for running level as opposed to forming definitions built by the panel may have influenced your conclusions.

21. Please expand on the results of your running level definitions in your "Discussion" section.

Reviewer #3: General

The paper is well written and the study uses appropriate methodology for reaching consensus regarding standards for classifying runners as well as for recommendations for running footwear.

One major concern that I have is that while the data was collected anonymously, the country and region of the country is provide din the raw data. This information along with the acknowledgment to specific participants, makes it quite easy to identify the responses of many of the participants in the raw data. The country and region data collected in the survey needs to be deleted to de-identify the data and preserve anonymity of the participants responses.

Another concern I have is the use of a manuscript in review as a major reference for this study. The Hoitz et al, manuscript that is listed as in review is not available to the reviewers of the current manuscript. As such it is difficult to discern how the current manuscript contributes to the literature. Moreover, depending on when or if the Hoitz, et al manuscript is accepted, it may not be available to the readers of the current manuscript. It would be acceptable to reference a manuscript that has been accepted and is in press.

Minor

Line 111: the phase “reached out to”, is awkward perhaps “contacted” or similar

Table 3 or discussion of runner classification. While consensus was reached on runner classification, was consensus reached on how to classify runners who may meet standards across categories (e.g. run at novice speed but with the habit or experience of recreational runners). For example, for a runner to be in a category do they have to meet 4 of the 5 categories or … ?

Table 6. I re-read the methods paragraph describing the manner of reaching consensus multiple times, lines 181-194. I also read the results paragraph regarding shoe properties, lines 283 to 293, multiple times. However, it is not clear to be which specific variables qualified to be presented in table 6.

6. PLOS authors have the option to publish the peer review history of their article (what does this mean?). If published, this will include your full peer review and any attached files.

Reviewer #1: Yes: Garrett Scott Bullock

Reviewer #2: No

Reviewer #3: No

---

## [Author Response · Author response to Decision Letter 0]

26 May 2020

See responses in green. (Please see our uploaded version for the differentiating color)

Reviewer #1: General Comments:

The reviewer would like to commend the authors for undertaking an important and interesting topic. Determining the shoe recommendations for different running levels is an important topic, that can aid clinicians and running coaches in choosing the right foot wear for different runners of different abilities.

Overall this is a well written manuscript, with good methodology. There are some specific comments which are written below.

>>Thank you for your compliments and suggestions. They have improved our manuscript.

Abstract:

General comment: For an abstract, the background should be brief. Suggest only have 2 sentence for the background. I do not think you need to describe why a Delphi study is powerful within the abstract. I think the first 3 sentences would suffice, and potentially reduce these three sentences into 2.

Within the abstract methods, a little bit more information is needed. For example, how many questions did the study begin with, and how were they whittled down through the three rounds, and how was data tallied. Further, within the results, you describe that there were originally 20 proposed variables. This is an example of something that needs to be in the methods.

>>We have updated the abstract as suggested with the following:

“Providing runners with footwear that match their functional needs has the potential to improve footwear comfort, enhance running performance, and reduce the risk of overuse injuries. It is currently not known how footwear research experts make decisions about different shoe features and their properties for runners of different levels. We performed a Delphi study in order to understand: 1) definitions of different runner levels, 2) which footwear features are considered important, and 3) how these features should be prescribed for runners of different levels. Experienced academics, journalists, coaches, bloggers and physicians that examine the effects of footwear on running were recruited to participate in three rounds of a Delphi study. Three runner level definitions were refined throughout this study based on expert feedback. Experts were also provided a list of 20 different footwear features. They were asked which features were important and what the properties of those features should be.” (line 26-35)

Need key words at the end of the abstract.

>>Thank you for the reminder. We have included the following key words: Individualized footwear, running biomechanics, runner abilities, footwear experts, midsole hardness

Introduction:

Line 54: Delete the parenthetical citation fully written citation, should just be a reference number.

>>This citation has been replaced with the appropriate number.

Line 67: Same here, please deleted written citation, should just be a reference number.

>>This citation has been replaced with the appropriate number.

Line 69-70: Reword this to not be a numbered list. Within the intro, it should just be written sentences.

>>We have removed the numbers from the sentence and updated the text to the following:

“On the other hand, there has been little scientific attention on footwear features such as outsole traction or forefoot flares which could indicate: the prescription of these features to different runner levels is trivial, or that these features are not considered important by footwear professionals, or little is known on how to prescribe these features.” (line 78-81)

Line 71: You state, “it is close to impossible for running footwear professionals to provide evidence-based recommendations for footwear properties for runners of different levels.” But then you go on to say you are performing a Delphi to find the best recommendations from the experts. I think this is contradictory. I think you should focus more on how there is not clarity on professional recommendations for footwear for different running skills or groups.

>>We have removed the last comment from the introduction to here as we addressed these two comments together. The corresponding statements now read: 

“In summary, there is a need to better understand how footwear research experts make decisions about different footwear features and their properties” (Lines 81-83)

I think the third to last and second to last paragraphs can be amalgamated into one paragraph. Further, the second to last paragraph ends abruptly and a better conclusion is need to set up the purpose paragraph.

>>We have combined the two paragraphs and updated the phrasing so that it is more focused on “how there is not clarity in professional recommendations”:

“Modern running shoes are complex systems. They incorporate many different features (e.g., crash-pads, heel counters, flares, midsole hardness) and each of these features can be included, excluded, and/or tuned individually to modify the characteristics of the final running shoe system (e.g., cushioning, stability, heel-to-toe transition, energy return). Some of these shoe features have been studied more extensively, such as rearfoot midsole hardness, while others have received little attention, such as upper breathability (12). Nevertheless, a strong research focus on certain footwear features (e.g., midsole hardness) does not necessarily translate into agreement on how modifying these features may affect the running mechanics, performance, injury risk, or footwear comfort in runners of different levels. For example, a recent review found inconclusive evidence regarding the biomechanical effects of different midsole hardness – one of the most studied footwear features (12). On the other hand, there has been little scientific attention on footwear features such as outsole traction or forefoot flares which could indicate: the prescription of these features to different runner levels is trivial, or that these features are not considered important by footwear professionals, or little is known on how to prescribe these features. In summary, there is a need to better understand how footwear research experts make decisions about different footwear features and their properties. A powerful way to examine these decisions is to gather and summarize opinions of experts in the field of running biomechanics and footwear using a Delphi study. The Delphi method has been utilized for gathering and summarizing opinions via survey-based responses of an expert panel in order to obtain consensus on complex topics. For example, this technique has been successfully applied to establish the now frequently reported “Minimalist Index” of running shoes (13). Such an understanding can target future systematic investigations around the presumed optimal property of important footwear features.” (Lines 69-88)

Line 73-76: Why are aims here and the purpose in the final introduction paragraph? This is confusing for the reader. Suggest only having the purpose at the last intro paragraph and deleting the aims.

>>We have deleted the aims as suggested.

Methods:

General comments: An overall study design sub section is needed at the beginning of the methods. This should give the 10,000 foot view of the study.

>>We have included an overview at the beginning of the Methods section: 

“Footwear research experts were asked to complete three rounds of a Delphi study, with each successive round building on the results gathered from the previous round. Three runner level definitions were refined throughout the three rounds of the Delphi study through expert feedback. Experts were also provided a list of 20 different footwear features. Through the three rounds of the study, experts provided opinions on which features are important and what the properties should be for the footwear features for the three different running levels.” (lines 96-101)

You need to give inclusion/exclusion criteria for who was considered an expert for this study.

>>We have added the following exclusion criteria in the methods:

“Participants were excluded if they had under two years of research experience related to running footwear.” (line 113-114)

Lines 103-117: I see that 142 experts were contacted. How many responded and were included. A flow chart might help the reader to understand this process.

>>We have included a consort diagram (new Figure 1) to show the number of experts in each round.

Line 122: Need to cite the running lit used. Further, it is confusing with the parenthetical statement “as detailed below. Suggest deleting this.

>>We have deleted the parenthetical statement and replaced it with the citations used in the subsequent paragraph (line 128).

Novice versus Recreational runner definition: In the novice group, you state that they run no more than 20km/week, but in the recreational group, they run 10-50 km/week? How do delineated between someone that runs 15-20 km/week? Is this based off of times per week (0-3 v 1-5)? Please clarify. High Caliber runners: I see the same thing here, they run 30km+/week. Please clarify

>>Thank you for clarifying this. We have included the following description to clarify the overlapping mileage:

“The proposed characteristics provide guidelines for runner classification. As such, there is overlap in the running distance per week between the different running levels in order to accommodate runners that train less and have a better running performance.” (line 128-130)

Line 211: Can you explain further why the ‘don’t know’ questions were not included in round 3?

>>We have expanded upon our explanation with the following:

“These questions were only included in the second-round as we received feedback from the experts that the questionnaire was time consuming which may have increased the drop out rate if these questions were asked in the third-round again.” (line 231-233)

Line 218-219: Please add the software program used to calculate these statistics.

>>We added the software program that we used to calculate the statistics in the “Analysis and Visualization” section: “All statistical analyses were performed in MATLAB (MathWorks, Natick, MA, USA)”.(line 240)

Results:

General comment: It is not recommended to use bullet points within the results. Please edit accordingly

>>We have eliminated the bullet points and updated the text to the following:

“The respondents’ rating of the running level definitions improved as the Delphi study progressed. The median score given to the running level definitions increased each round and the interquartile range decreased as 88% of respondents rated the running level definitions between 7 and 10 in the third-round as opposed to 69% in the first-round (see Fig. 3). The changes to the running level definitions for the second-round were: increased “novice” running experience to one year (from six months) and increased “recreational” running experience to greater than one year (from six months), increased “high caliber” running habits to >4 sessions/week (from >3 sessions/week) and >50 km/week (from >30 km/week), specified the running performance as males between the ages of 18 to 34, replaced “stress management” with enjoyment for running motivation for all levels, re-order the “high caliber” running motivation from 1) Improve general health, 2) Stress management, 3) Competition to: 1) Competition, 2) Improve general heath, and 3) Enjoyment, and re-order the priorities for footwear design for “High caliber” from: 1) Improve performance, 2) Improve comfort, 3) Reduce injury risk, to: 1) Improve performance, 2) Reduce injury risk, 3) Improve comfort. Subsequent changes to the running level definitions were to ensure that the high caliber and recreational runner 5km and 10 km time were indicative of the respective marathon times. These updates resulted in the final updated runner level definitions in Table 3.” (line 251-266)

Overall the results are well written.

>>Thank you!

Discussion:

Line 307-10: These are more article strengths and should be moved to the strength and limitations section, not the summary discussion paragraph.

>>We have eliminated the sentences in question.

Line 315-17: This is a future research, implications, and/or conclusion sentence and should be moved

>>I understand that this concluding sentence pertains to future research, in our opinion it is a major point of discussion. We have kept this sentence as it wraps together the discussion summary paragraph.

Line 369: Suggest deleting the term ground truth and just state that this should serve as valuable information, etc.

>>We have eliminated “ground truth” and updated the sentence to the following:

“As such, the findings from this study can serve as a valuable starting point for future systematic biomechanical investigations examining the influence of footwear features on runners with different training/performance levels.” (line 412-414)

Line 373-381: suggest that this paragraph be moved before the limitations paragraph.

>>We have moved the paragraph before the Limitation paragraph as suggested. (line 379-387)

Line 384-386: Delete the first sentence, this can be said in the strengths paragraph but the conclusion paragraph should focus on the findings and future directions.

>>We have removed the strengths portion of the sentence and updated it to:

“Footwear research experts provided feedback on the effects of different footwear features on running biomechanics across three running levels as well as provided a consensus on the characteristics of runners in these different running levels.” (line 418-427)

Reviewer #2: Overall this manuscript fills an obvious void in the literature and aims to assist researchers, clinicians, coaches, and running enthusiasts with shoe prescriptions, while also informing future running shoe research. This work is generally well written and free from fundamental flaws; however, several minor revisions to the proposed article will undoubtedly improve this already great work.

>>Thank you for your kind comments. Your suggestions have improved the manuscript. 

1. The words "the participants" are over utilized throughout the manuscript. Varied diction will help to maintain reader interest and attention.

>>We have updated the manuscript so that there is more varied diction.

2. As this is a study employing Delphi techniques no statistical analyses are necessary and furthermore, no analyses were actually conducted. The "Statistical Analysis" section is therefore unnecessary and the subsequent descriptive statistics can simply be presented in the "Results" as well as Fig 3.

>> Together with your suggestion and the Reviewer #1’s comment about what program the statistics were performed in and your #11 comment, we have updated the “Statistical Analysis” section to “Analysis and Visualization” section.

3. A more clear and consistent distinction between footwear properties and features throughout the manuscript would improve readability.

>>We have checked the entire manuscript and ensured that “property” and “feature” were used correctly. 

4. "Appendix A" utilizes the term "categories" as opposed to "properties" further illustrating the previous point.

>>We have updated “categories” to “Property categories” throughout the Appendix and updated the file name of Appendix A to: “S1 Appendix A – Shoe Feature Descriptions and Properties”

5. Additional headings for the "Footwear Properties" in the "Methods" and "Results" sections would assist readers navigating between parts of the manuscript.

>>We have added sections titled: “Footwear Feature Properties” in both the Methods and Results sections.

6. The described methods for determining footwear features and feature properties importance is challenging to read at times (particularly lines 169-179; lines 188-192); please try to concisely and succinctly explain these steps.

>>We have updated the mentioned sections with the following:

“The importance of the footwear features was assessed in the first-round and verified in the second-round. In the first-round, participants were asked if footwear features are important when designing footwear for different running levels. The experts could choose between the following for each footwear feature: (a) is important, (b) is not important or (c) they do not know if it is important. The footwear features were important if over 75% of the first-round participants selected option (a). Prior Delphi studies have defined consensus between 51% (21) and 80% (22) of respondents. The important features were then presented to the second-round participants. The participants were asked if they agreed with each of the features selected as important/non important on a 10-point scale where “1” indicated that the list of important/non important features were “Not at all appropriate” and “10” indicated “Most Appropriate”. The list of important features was verified if over 75% of the second-round participants answered with a seven or higher on the 10 point-scale. The second- and third-rounds of the Delphi study were then limited to the important footwear features. In each round, the experts were asked if other footwear features should be included in the Delphi study. If there were at least five suggestions to add a certain feature, this new footwear feature was added to the subsequent round and the participants were asked if the newly added footwear features were important.” (lines 187-201)

and

“The experts were asked to recommend footwear feature properties for the different running levels in each round of the study from a multiple-choice selection (see Appendix A for the lists of footwear feature properties). Most footwear feature properties were obtained through literature; however, if there was no related literature (e.g., upper elasticity), properties were provided based on commercially available shoes. In rounds 2 and 3, the results from the previous round were presented to the participants. If at least 51% of the participants agreed on a footwear feature property for a specific running level (e.g., high breathability for novice runners), the participants would be asked if they agreed with the consensus the next round. If at least 51% of the participants verified the consensus, the experts were not asked again to recommend a footwear feature property for that running level (see Fig. 2). In comparison to the consensus for the importance of shoe features (agreement of 75% of respondents), the threshold for consensus was set lower for agreement on footwear feature properties (51%) because of the greater number of available response options.” (lines 204-215)

7. Lines 181-184 seem somewhat redundant.

>>We have removed the sentence.

8. The reference to Fig 2 in line 182 seems somewhat premature. Describing the general flow of these methods prior to interpreting Fig 2 made this section easier for this reviewer to understand.

>>We have moved the reference to Fig. 2 to near the end of the paragraph after the explanation of the methods. 

9. It is not clear how the Likert scale used to rate footwear features (as described in the "Methods" section) is actually used in this study.

>>We have clarified the use of the Likert scale by including the following in our methods:

“The list of important features was verified if over 75% of the second-round participants answered with a seven or higher on the 10 point-scale.” (line 196-197)

10. Fig 2 is very helpful, but a threshold of >50% is provided when the text describes using a 51% threshold.

>>We have updated the Fig. 2 and replaced “>50%” with “≥51%”.

11. While minor, the software used to produce images was not stated.

>>We have added the following in methods: “Figures were created in MATLAB and Adobe Illustrator (San Jose, CA, USA).” (line 220-241)

12. Line 162 - Explicitly cite why/where the 20 features considered comes from.

>>We have included the following to describe how we came to the 20 footwear features:

“. These 20 features were chosen from a list of 31 running shoe footwear features that were identified based on an initial literature review, market analysis, and internal discussion. Two influential studies during this process were reports from (6) and (13). The initial list of 31 was reduced to 23 features by removing or joining related features that were reflected in other features or similar in their function, respectively (e.g., remove midfoot midsole hardness and only retain forefoot and rearfoot midsole hardness). Pilot testing with four footwear science experts (not included in the main study) indicated that 23 features resulted in a questionnaire that would require more than an hour to complete and could potentially lead to a high-drop out rate. Therefore, we limited the number of footwear features to 20, by removing features for which pilot participants indicated low relevance (e.g. upper overlays or varus alignment). In return, the option was added for experts of the main study to suggest footwear features, that should be added to the questionnaire.” (line 169-180)

13. The inclusion of 2 aims and 3 purposes is somewhat confusing. I recommend removing the aims from your "Introduction" as they do not match the "Methods" and "Results" sections as obviously.

>>We have removed the two aims from the introduction. 

14. Please ensure that permissions for any adapted images (i.e. Figs 1 & 4) are provided as necessary.

>>Figures 1 and 4 have been removed as we have replaced the Hoitz article (currently still in review) with another recent running shoe construction review paper (Sun et al., 2020). “Sun, X, Lam, WK, Zhang X., Wang J, & Fu W (2020). Systematic Review of the Role of Footwear Constructions in Running Biomechanics: Implications for Running-Related Injury and Performance. Journal of Sports Science and Medicine,19, 20-37”

15. A limitation that seems somewhat overlooked is that the definitions of runner levels changed throughout iterations. As these definitions changed, so too may have respondents' recommended properties. While the 3 repetitions and consensus measures may help to quell these concerns, it seems important to consider the implications of these interconnected moving targets.

>>We have added the following to the limitations as you suggested:

“The recommended footwear feature properties may have been influenced by a dynamic definition of the runner levels, which changed slightly throughout the study. These changing definitions, however, seemed to have little effect on expert opinions on the footwear feature properties as the verifying consensus level was generally higher than the original consensus level (Table 4, last vs. second-to-last column).” (line 403-408)

16. If possible, I would like to know more about your "Additional Delphi Questions" results in the discussion. I read some of the statements in your raw data set and found the additional insights very compelling. You do a good job of introducing some of the identified themes in your "Discussion" but I feel that a bit more would elevate the current manuscript.

>>We have integrated some expert feedback in the running level definitions discussion paragraph (line 434-451). Please see our response below #20 and #21.

17. Tables 5 and 6 both seem to provide complementary results. Is there a way to combine them or make the more exclusive from one another?

>>We have eliminated Table 6 and added a column for “% Participant in agreement with consensus” to Table 5. 

18. Consider a CONSORT diagram so readers can better understand the development of the expert panel round by round.

>>We have included a consort diagram, as suggested, to show the number of experts in each round as the new Figure 1.

19. Please expand on how your panel may or may not influence your conclusions in the "Discussion" (e.g. Where they all from the US? Do they disproportionately represent companies with financial interests in designing complicated shoes? Etc.).

>>We have included the following to expand on our panel in the limitation:

“Furthermore, the final recommendation may have been biased as more experts that completed the survey were male (e.g., 22/26 of the final participants). This expert panel was otherwise diverse as nine countries were represented.” (line 401-403)

20. Please discuss how providing the expert panel with definitions in round 1 for running level as opposed to forming definitions built by the panel may have influenced your conclusions.

21. Please expand on the results of your running level definitions in your "Discussion" section.

>>We have expanded our running level definitions discussion that includes discussion of #20:

“The footwear experts came to a consensus on the running level definitions through slight adjustments to the initial definitions proposed and derived from literature. We opted to provide initial running level definitions to our expert panel rather than letting the panel formulate definitions independently. This latter approach would have required additional Delphi rounds prior to the recommendation of footwear features and their properties. Panel formulated definitions may have resulted in different running level definitions compared to the approach presented here and different running level definitions could have led to altered footwear feature recommendations. However, the experts’ consensus on the running level definitions were in agreement with prior literature. This is exhibited by the novice runner level definition which is similar to a definition created based on subjective running questionnaires (7). The experts did recommend an increased workload for high caliber runners in comparison to literature (7) as participant feedback resulted in the distance per week to be increased from >30 km/week to >50 km/week. These definitions may be viewed more as guidelines as one footwear expert mentioned that “Even elite athletes perform training runs with different intensities, durations, on different surfaces and so on. For each of these runs they might select a different type of footwear.” This comment touches on the competing requirements for running shoes and there may be multiple “correct” shoes for a given running level, especially in the high caliber category.” (lines 362-377)

 

Reviewer #3: General

The paper is well written and the study uses appropriate methodology for reaching consensus regarding standards for classifying runners as well as for recommendations for running footwear.

>>Thank you for your compliments and suggestions.

One major concern that I have is that while the data was collected anonymously, the country and region of the country is provide din the raw data. This information along with the acknowledgment to specific participants, makes it quite easy to identify the responses of many of the participants in the raw data. The country and region data collected in the survey needs to be deleted to de-identify the data and preserve anonymity of the participants responses.

>>We have de-identified the raw data by removing the country and region for each participant. 

Another concern I have is the use of a manuscript in review as a major reference for this study. The Hoitz et al, manuscript that is listed as in review is not available to the reviewers of the current manuscript. As such it is difficult to discern how the current manuscript contributes to the literature. Moreover, depending on when or if the Hoitz, et al manuscript is accepted, it may not be available to the readers of the current manuscript. It would be acceptable to reference a manuscript that has been accepted and is in press.

>> We have removed the citation in question (as the mentioned manuscript is still in review) and replaced it with the following: Sun, X, Lam, WK, Zhang X., Wang J, & Fu W (2020). Systematic Review of the Role of Footwear Constructions in Running Biomechanics: Implications for Running-Related Injury and Performance. Journal of Sports Science and Medicine,19, 20-37

Minor

Line 111: the phase “reached out to”, is awkward perhaps “contacted” or similar

>>We have updated the phrasing as recommended. (line 112)

Table 3 or discussion of runner classification. While consensus was reached on runner classification, was consensus reached on how to classify runners who may meet standards across categories (e.g. run at novice speed but with the habit or experience of recreational runners). For example, for a runner to be in a category do they have to meet 4 of the 5 categories or … ?

>>While we did not specify how many criteria had to be fulfilled in order to decide the runner’s category at the beginning of the survey, we acknowledge your points and added the following to the limitation section:

“A limitation of the consensus process for the running level definitions was that we did not specify to the experts how many of the of the categories a runner must match to be considered a “novice”, “recreational”, or “high caliber” runner. As such, the definitions may lead to minor variations when different footwear experts categorize runners.” (lines 408-411)

Table 6. I re-read the methods paragraph describing the manner of reaching consensus multiple times, lines 181-194. I also read the results paragraph regarding shoe properties, lines 283 to 293, multiple times. However, it is not clear to be which specific variables qualified to be presented in table 6.

>>We have eliminated Table 6 and added the “% Participants in agreement with consensus” column from Table 6 to Table 5.

---

## [Decision Letter · Decision Letter 1]

10 Jun 2020

PONE-D-20-07118R1

Shoe Feature Recommendations for Different Running Levels: A Delphi Study

PLOS ONE

Dear Dr. Honert,

Thank you for submitting your manuscript to PLOS ONE. After careful consideration, we feel that it has merit but does not fully meet PLOS ONE’s publication criteria as it currently stands. Therefore, we invite you to submit a revised version of the manuscript that addresses the points raised during the review process.

After careful review, I believe reviewer 2 has noted several areas that can strengthen your paper. As many of those are simple errors to correct or streamlining text, this should not be difficult. I would also ask you to carefully consider comments noted at lines **82, 96, 113, 117, 191-192, 240, and 323. **Please do you best to address these issues. I look forward to seeing your revisions.

We look forward to receiving your revised manuscript.

Kind regards,

Chris Harnish, PhD

Academic Editor

PLOS ONE

Reviewers' comments:

Reviewer's Responses to Questions

**Comments to the Author**

1. If the authors have adequately addressed your comments raised in a previous round of review and you feel that this manuscript is now acceptable for publication, you may indicate that here to bypass the “Comments to the Author” section, enter your conflict of interest statement in the “Confidential to Editor” section, and submit your "Accept" recommendation.

Reviewer #1: All comments have been addressed

Reviewer #2: (No Response)

2. Is the manuscript technically sound, and do the data support the conclusions?

Reviewer #1: Yes

Reviewer #2: Yes

3. Has the statistical analysis been performed appropriately and rigorously? 

Reviewer #1: Yes

Reviewer #2: N/A

4. Have the authors made all data underlying the findings in their manuscript fully available?

Reviewer #1: Yes

Reviewer #2: Yes

5. Is the manuscript presented in an intelligible fashion and written in standard English?

Reviewer #1: Yes

Reviewer #2: Yes

6. Review Comments to the Author

Reviewer #1: Thank you for addressing all reviewer comments. I have no other reviewer editorial suggestions for this paper.

Overall Good work on this paper. It is interesting and pertinent.

Reviewer #2: While many (if not most) of my original comments were addressed by the authors, several small problems still exist in my opinion. Similar to my previous assessment, these minor points negatively influence the readability and overall impact of this important work. I believe that careful attention to these points listed below will improve the manuscript.

Ensure that figures and tables (specifically table footnotes and legends) are properly formatted.

Ensure citations are properly and consistently formatted (see 41 and 42 for examples).

Ensure double spacing throughout the text and eliminate unnecessary spacing.

Line 82 - The term "footwear research experts" seems somewhat inappropriate as active researchers were not explicitly recruited and may also only make up a portion of your participants. Consider simplifying by using the term "footwear experts." Even though experts are often involved with or active in conducting research, they do not necessarily need to participate in research to be labeled as experts.

Line 82 - Use of the word "powerful" seems subjective. Furthermore, this sentence seems somewhat redundant given the following line describing the successful use of a Delphi study in the field. Consider combining these 2 statements and using less conjecture.

Line 96 - Again, the use of the term "footwear research experts" does not likely accurately describe the entire population of experts recruited.

Lines 106 & 123 - Figure 1 should be moved to, and initially referenced in, the "Results: Participation" section.

Line 113 - It is stated that individuals without at least 2 years of research experience were excluded from participation. As stated, this criteria seems to have been ignored. The definition of "research," especially within the context of a peer-reviewed journal, seems likely to exclude a majority of the persons tapped to participate, as well as a number of individuals that may have participated in the study. This point needs to be clarified or revised.

Line 117 - "and were able to provide feedback..." The use of the word "able" is somewhat misleading given the dual nature of the word in context. The reader could interpret that all round 1 participants continued through rounds 2 and 3; or the reader could think that all participants were invited to again participant in subsequent rounds. Additionally, if an individual did participate in say round 1 but did not in round 2, were they invited back for round 3? This broader question should be more explicitly detailed (lines 119-121 seem to vaguely describe this question).

Line 188 - Tense shift "are." The question as stated does not broadly apply to each feature as I believe the authors intend. Consider: "In the first-round, participants were asked if [each] footwear features [was] important when designing..."

Lines 190-191 - Restructure the sentence using basic "if - then" logic to simplify reading.

Lines 191-192 - This sentence should likely come before the prior one as it establishes a standard for your selection criteria; however, additional information is also necessary to justify why 51% or 80% thresholds were not used.

Line 206 - "Most footwear feature properties were obtained through literature..." What literature: "the literature," "a review of the literature," etc? More information similar to the features list is necessary.

Line 240 - Statistical analysis was not performed. Descriptive statistics are not analytic statistics.

Lines 240-241 - What version of MATLAB and Illustrator were used?

Table 2 - The title is too long and complicated. An appropriate title should be short and briefly describe the global goal(s) of the table. Any additional or necessary information should be described as a footnote. Additionally a footnote should be added to reflect the transition of 1 participant from one group to another.

Lines 254-263 - This sentence is very difficult to read and covers too much material.

Tables 4 & 5 - The titles are too long and complicated. An appropriate title should be short and briefly describe the global goal(s) of each table. Any additional or necessary information should be described in footnotes. Both tables also inconsistently use the term "participants" and "respondents" to describe the expert panel. This is confusing and should be consistently reported one way or the other. Tables 4 & 5 also cover very similar material; however, their presentation is very different from one another. Consider combining or revising these tables further to enhance readability.

Table 5 - Putting the "*" next to the percentage seems somewhat redundant. Consider putting this next to the property running level.

Line 323 - I take pause again with the use of the word "researching" here to broadly describe your population of experts. No clear attempts to recruit ONLY those actively involved with footwear research were described by the authors. Even though experts are often involved with or in research, they do not necessarily need to be in order to be labeled as experts.

Line 333 - Instead of the word "topics" consider using "features" to clearly identify what has been studied. Additionally, "i.e." may have been used incorrectly here. Consider revising according to your intended list of these features (if you are only listing and describing only 2 features, then a colon is more appropriate). (ALSO see the next statement).

Line 334 - This sentence should not begin with the citation ["(12)"]. Identify the author(s) according to the reference guidelines and use the numeric citation at the end. The use of the word "topics" is somewhat vague, consider "features" as that is what you are referencing. Also consider combining this sentence with the previous as they are saying the same thing.

Lines 335-337 - This sentence seems to assume that the listed publications were described in the previous sentence(s) through the use of the word "these;" however none of these works were described or referenced in regard to the point being made. Please reorganize this sentence, and/or the previous sentences, to ensure readability.

Line 417 - The term "footwear research experts" is a somewhat counterintuitive term to use given that researchers make up only a portion of your participants as well as your audience. Consider simplifying by using the term "footwear experts."

Lines 423-424 - The word "research" should be in the past tense. This sentence also seems somewhat empty given that 2 features are identified earlier in the "Discussion."

An additional limitation that needs to be addressed is the transferability of these results to various running surfaces .

7. PLOS authors have the option to publish the peer review history of their article (what does this mean?). If published, this will include your full peer review and any attached files.

Reviewer #1: Yes: Garrett Scott Bullock

Reviewer #2: No

---

## [Author Response · Author response to Decision Letter 1]

22 Jun 2020

>> We want to thank both of the reviewers for their additional time in reviewing our manuscript. We want to let the reviewers know that we have added in the discussion figure that was removed after the first-round of reviews. We have added this figure again as it provides a more compelling case as to which important footwear features should be researched in the future. This figure utilizes data, with permission, from Hoitz et al., 2020 (recently published: link). Please see uploaded version for the link and the colored responses.

Reviewer #1: Thank you for addressing all reviewer comments. I have no other reviewer editorial suggestions for this paper.

Overall Good work on this paper. It is interesting and pertinent.

>>Thank you for you compliments. 

Reviewer #2: While many (if not most) of my original comments were addressed by the authors, several small problems still exist in my opinion. Similar to my previous assessment, these minor points negatively influence the readability and overall impact of this important work. I believe that careful attention to these points listed below will improve the manuscript.

>>Thank you for your additional comments, which have improved our manuscript. Similar and relevant comments have been grouped so that they can be addressed together.

Ensure that figures and tables (specifically table footnotes and legends) are properly formatted.

>> We have updated the figures and tables according to PLOS standards, including font type used in figures, separating the figure captions and legends, placing the table captions below the tables, and succinctly describing the figures/tables.

Table 2 - The title is too long and complicated. An appropriate title should be short and briefly describe the global goal(s) of the table. Any additional or necessary information should be described as a footnote. Additionally a footnote should be added to reflect the transition of 1 participant from one group to another.

>> We have updated the title of Table 2 to the following: “Number of participants and their experience investigating/designing footwear.”

>> We have updated the caption of Table 2 to the following: “Note that one academic moved to industry between the first and second rounds of this study.”

Tables 4 & 5 - The titles are too long and complicated. An appropriate title should be short and briefly describe the global goal(s) of each table. Any additional or necessary information should be described in footnotes. Both tables also inconsistently use the term "participants" and "respondents" to describe the expert panel. This is confusing and should be consistently reported one way or the other. Tables 4 & 5 also cover very similar material; however, their presentation is very different from one another. Consider combining or revising these tables further to enhance readability.

>>We have updated “respondents” to “participants” in Table 4.

>>We have updated the title of table 4 to: “Percent of participants that agreed upon the importance of shoe features.”

>>We have updated the title of table 5 to: “Shoe feature properties that was most frequently chosen for each running level.”

Table 5 - Putting the "*" next to the percentage seems somewhat redundant. Consider putting this next to the property running level.

>>We have removed the asterisk from Table 5.

Ensure citations are properly and consistently formatted (see 41 and 42 for examples).

>> Thank you for bringing this to attention. We have updated the citations mentioned (see below) and checked all other citations.

“41. Tilp M. Benno M. Nigg, Maurice M. Mohr & Sandro R. Nigg – New paradigms in running injury prevention. Curr Issues Sport Sci. 2019 May 7;4(100). 

42. Oh K, Park S. The bending stiffness of shoes is beneficial to running energetics if it does not disturb the natural MTP joint flexion. J Biomech. 2017 Jan 18;533:127–35.”

Ensure double spacing throughout the text and eliminate unnecessary spacing.

>> We have eliminated unnecessary spacing and formatted table/figure captions to double spacing.

Line 82 - The term "footwear research experts" seems somewhat inappropriate as active researchers were not explicitly recruited and may also only make up a portion of your participants. Consider simplifying by using the term "footwear experts." Even though experts are often involved with or active in conducting research, they do not necessarily need to participate in research to be labeled as experts.

Line 96 - Again, the use of the term "footwear research experts" does not likely accurately describe the entire population of experts recruited.

Line 417 - The term "footwear research experts" is a somewhat counterintuitive term to use given that researchers make up only a portion of your participants as well as your audience. Consider simplifying by using the term "footwear experts."

>> We have updated “footwear research experts” to “footwear experts” throughout the manuscript.

Line 82 - Use of the word "powerful" seems subjective. Furthermore, this sentence seems somewhat redundant given the following line describing the successful use of a Delphi study in the field. Consider combining these 2 statements and using less conjecture.

>>We have combined the two sentences as recommended:

“An understanding of how footwear experts make decisions about different footwear features and their properties can be obtained through gathering and summarizing opinions of experts in the field of running biomechanics and footwear using a Delphi study.”

Lines 106 & 123 - Figure 1 should be moved to, and initially referenced in, the "Results: Participation" section.

>>We have moved Fig. 1 into the results section.

Line 113 - It is stated that individuals without at least 2 years of research experience were excluded from participation. As stated, this criteria seems to have been ignored. The definition of "research," especially within the context of a peer-reviewed journal, seems likely to exclude a majority of the persons tapped to participate, as well as a number of individuals that may have participated in the study. This point needs to be clarified or revised.

>>We have removed the word “research” and have update the sentence to:

“Participants were excluded if they had under two years of experience related to running footwear in their respective fields of expertise”

Line 117 - "and were able to provide feedback..." The use of the word "able" is somewhat misleading given the dual nature of the word in context. The reader could interpret that all round 1 participants continued through rounds 2 and 3; or the reader could think that all participants were invited to again participant in subsequent rounds. Additionally, if an individual did participate in say round 1 but did not in round 2, were they invited back for round 3? This broader question should be more explicitly detailed (lines 119-121 seem to vaguely describe this question).

>> We have removed the word “able” and clarified the text as to the participation with the following:

“The footwear experts completed web-based surveys through QuestionPro (questionpro.com) and could provide feedback after the completion of each round of this Delphi study. The participants that completed the first-round survey were invited to participate in the second-round. Similarly, the participants that completed the second-round survey were invited to participate in the third round.”

Line 188 - Tense shift "are." The question as stated does not broadly apply to each feature as I believe the authors intend. Consider: "In the first-round, participants were asked if [each] footwear features [was] important when designing..."

>> We have updated the text as recommended.

Lines 190-191 - Restructure the sentence using basic "if - then" logic to simplify reading.

Lines 191-192 - This sentence should likely come before the prior one as it establishes a standard for your selection criteria; however, additional information is also necessary to justify why 51% or 80% thresholds were not used.

>>We have updated the text to the following:

“If over 75% [a similar threshold to (22,23)] of the first-round participants selected option (a), the footwear feature was defined as important.”

>>We have included the following citations to support the selection of 75% threshold:

22. Cook C, Brismée J-M, Fleming R, Sizer PS. Identifiers Suggestive of Clinical Cervical Spine Instability: A Delphi Study of Physical Therapists. Phys Ther. 2005 Sep 1;85(9):895–906. 

23. Binkley J, Finch E, Hall J, Black T, Gowland C. Diagnostic Classification of Patients with Low Back Pain: Report on a Survey of Physical Therapy Experts. Phys Ther. 1993 Mar 1;73(3):138–50. 

Line 206 - "Most footwear feature properties were obtained through literature..." What literature: "the literature," "a review of the literature," etc? More information similar to the features list is necessary.

>>This information is within S1 Appendix. We have included a call-out in the sentence and reads as such:

“Most footwear feature properties were defined based on the reviewed footwear literature (see S1 Appendix);”

Line 240 - Statistical analysis was not performed. Descriptive statistics are not analytic statistics.

>> We have added the following to the statistics portion of our manuscript: 

“Paired statistical analyses were performed to determine if the running level definitions improved through the three rounds of this Delphi study. A Friedman’s test was performed utilizing the subjective ratings from the respondents that participated in all three rounds of the study (N = 24). If the Friedman’s test revealed a significant effect, follow-up Wilcoxon signed-rank tests with a Bonferroni correction were performed to investigate pairwise differences between the individual rounds. The significance level α was set to 0.05 for all statistical tests.”

>>And the following to the results section:

“The respondents’ rating of the running level definitions improved as the Delphi study progressed, χ2 (2, N=24) = 13.95, p=0.0009.”

and

“The increase in the running level scores between the first and third rounds was statistically significant (p=0.006).”

Lines 240-241 - What version of MATLAB and Illustrator were used?

>>We have updated the text to include version numbers:

“All analyses were performed in MATLAB (version 2019a, MathWorks, Natick, MA, USA). Figures were created in MATLAB and Adobe Illustrator (version 22.1, San Jose, CA, USA).”

Lines 254-263 - This sentence is very difficult to read and covers too much material.

>>We have broken up the mentioned sentence and now reads as such:

“The increased running level ratings were accompanied by changes to the running level definitions. The changes to the “novice” running level definition for the second-round were: increased running experience to one year and replaced “stress management” with “enjoyment” for running motivation. The changes to the “recreational” running level definition for the second-round were: increased running experience to greater than one year and replaced “stress management” with “enjoyment” for running motivation. The changes to the “high-caliber” running level definition for the second-round were: increased running habits to >4 sessions/week and >50 km/week, replaced “stress management” with “enjoyment” for running motivation, re-order the running motivation to 1) Competition, 2) Improve general heath, and 3) Enjoyment, and re-order the priorities for footwear design to 1) Improve performance, 2) Reduce injury risk, 3) Improve comfort. We also specified the running performance as males between the ages of 18 to 34.”

Line 323 - I take pause again with the use of the word "researching" here to broadly describe your population of experts. No clear attempts to recruit ONLY those actively involved with footwear research were described by the authors. Even though experts are often involved with or in research, they do not necessarily need to be in order to be labeled as experts.

>>We have replaced “researching” to “examining”

Line 333 - Instead of the word "topics" consider using "features" to clearly identify what has been studied. Additionally, "i.e." may have been used incorrectly here. Consider revising according to your intended list of these features (if you are only listing and describing only 2 features, then a colon is more appropriate). (ALSO see the next statement).

>>We have revised the sentence as recommended.

Line 334 - This sentence should not begin with the citation ["(12)"]. Identify the author(s) according to the reference guidelines and use the numeric citation at the end. The use of the word "topics" is somewhat vague, consider "features" as that is what you are referencing. Also consider combining this sentence with the previous as they are saying the same thing.

>> We have eliminated the sentence in question and implemented the citation at the end of the topic sentence.

Lines 335-337 - This sentence seems to assume that the listed publications were described in the previous sentence(s) through the use of the word "these;" however none of these works were described or referenced in regard to the point being made. Please reorganize this sentence, and/or the previous sentences, to ensure readability.

>>We have updated the corresponding sentence to the following:

“Previous research has shown that a softer rearfoot midsole can reduce ground reaction force loading metrics such as vertical loading rate or peak impact forces (25–27), which have been hypothesized to reduce running-related injuries (28,29).”

Lines 423-424 - The word "research" should be in the past tense. This sentence also seems somewhat empty given that 2 features are identified earlier in the "Discussion."

>>We have updated the tense to “researched”

An additional limitation that needs to be addressed is the transferability of these results to various running surfaces.

>>We have added the following to the limitations section:

“These results may also not be generalizable to different running surfaces/terrains as we asked participants to only consider running on a hard surface.”

---

## [Decision Letter · Decision Letter 2]

29 Jun 2020

Shoe Feature Recommendations for Different Running Levels: A Delphi Study

PONE-D-20-07118R2

Dear Dr. Honert,

Thanks you for your hard work on revising this manuscript. We’re pleased to inform you that your manuscript has been judged scientifically suitable for publication and will be formally accepted for publication once it meets all outstanding technical requirements.

Kind regards,

Chris Harnish, PhD

Academic Editor

PLOS ONE

---

## [Editor Report · Acceptance letter]

6 Jul 2020

PONE-D-20-07118R2 

Shoe Feature Recommendations for Different Running Levels: A Delphi Study 

Dear Dr. Honert:

I'm pleased to inform you that your manuscript has been deemed suitable for publication in PLOS ONE. Congratulations! Your manuscript is now with our production department. 

Kind regards, 

on behalf of

Dr. Chris Harnish 

Academic Editor

PLOS ONE